# Targeted dimensionality reduction enables reliable estimation of neural population coding accuracy from trial-limited data

Charles R. Heller[ID][1,2¤], Stephen V. David[2]*

**1** Neuroscience Graduate Program, Oregon Health and Science University, Portland, Oregon, United States of America, **2** Oregon Hearing Research Center, Oregon Health and Science University, Portland, Oregon, United States of America

¤ Current address: Max Planck Institute for Biological Cybernetics, Tübingen, Germany
* davids@ohsu.edu

**Data Availability Statement:** All data and code used for producing the figures and analysis in the manuscript is available on GitHub at https://github.com/crheller/dDR. We have also used Zenodo to

## Abstract

Rapidly developing technology for large scale neural recordings has allowed researchers to measure the activity of hundreds to thousands of neurons at single cell resolution *in vivo*. Neural decoding analyses are a widely used tool used for investigating what information is represented in this complex, high-dimensional neural population activity. Most population decoding methods assume that correlated activity between neurons has been estimated accurately. In practice, this requires large amounts of data, both across observations and across neurons. Unfortunately, most experiments are fundamentally constrained by practical variables that limit the number of times the neural population can be observed under a single stimulus and/or behavior condition. Therefore, new analytical tools are required to study neural population coding while taking into account these limitations. Here, we present a simple and interpretable method for dimensionality reduction that allows neural decoding metrics to be calculated reliably, even when experimental trial numbers are limited. We illustrate the method using simulations and compare its performance to standard approaches for dimensionality reduction and decoding by applying it to single-unit electrophysiological data collected from auditory cortex.

## Introduction

Neural decoding analysis identifies components of neural activity that carry information about the external world (*e.g.* stimulus identity). This approach can offer important insights into how and where information is encoded in the brain. For example, classic work by Britten et al. demonstrated that the ability of single neurons in area MT to decode visual stimuli closely corresponds to animal's perceptual performance [1]. Thus, by using decoding the authors identified a possible neural substrate for detection of motion direction [1]. Yet, behavior does not depend solely on single neurons. In the years since this work, many theoretical frameworks have been proposed for how information might be pooled across individual neurons into a population code [2–8]. One clear theme that has emerged from this work is that stimulus

assign a DOI to the repository: 10.5281/zenodo.
5788573.

**Funding:** This work was supported by a National
Science Foundation Graduate Research Fellowship
(NSF GRFP, GVPRS0015A2) (CRH), the National
Institute of Health (NIH, R01 DC0495) (SVD),
Achievement Rewards for College Scientists
(ARCS) Portland chapter (CRH), and by the Tartar
Trust at Oregon Health and Science University
(CRH). The funders had no role in study design,
data collection and analysis, decision to publish, or
preparation of the manuscript.

**Competing interests:** The authors have declared
that no competing interests exist.

independent, correlated activity (*i.e.* noise correlations) between neurons may substantially impact information coding [2, 4–8]. This has now been confirmed *in vivo* using decoding analysis to measure the information content of large neural populations [9–11]. Therefore, covariability between neurons must be taken into account when measuring population coding accuracy.

Under most experimental conditions, estimates of pairwise correlation between neurons is unreliable due to insufficient sampling (*e.g.* too few stimulus repeats) [12]. In these situations, traditional decoding algorithms are likely to over-fit to noise in the neural data. This issue becomes even more apparent as the number of pairwise interactions that must be estimated increases, a situation that is becoming more common due to the recent growth in large-scale neurophysiology techniques [13]. In some cases, *e.g.* for chronic recording experiments and anesthetized preps, the number of trials can be increased to circumvent this issue. However, in behavioral experiments, where the number of trials is often fundamentally limited by variables such as animal performance, new analytical techniques for decoding are required.

Here, we present decoding-based dimensionality reduction (*dDR*), a simple and generalizable method for dimensionality reduction that significantly mitigates issues around estimating correlated variability in experiments with a relatively low ratio of observations to neurons. Our method takes advantage of recent observations that population covariability is often low-dimensional [14–17] to define a subspace where decoding analysis can be performed reliably while still preserving the dominant mode(s) of population covariability. The *dDR* method can be applied to data collected across many different stimulus and/or behavior conditions, making it a flexible tool for analyzing a wide range of experimental data.

We motivate the requirement for dimensionality reduction by illustrating how estimates of a popular information decoding metric, $d'^2$ [4, 5], can be biased by small experimental sample sizes. Building on a simple two-neuron example, we demonstrate that low-dimensional structure in the covariability of simulated neural activity can be leveraged to reliably decode stimulus information, even when the number of neurons exceeds the number of experimental observations. Finally, we use a dataset collected from primary auditory cortex to highlight the advantages of using *dDR* for neural population decoding over standard principal component analysis.

## Materials and methods

### Surgical procedure

All procedures were performed in accordance with the Oregon Health and Science University Institutional Animal Care and Use Committee (IACUC) and conform to standards of the Association for Assessment and Accreditation of Laboratory Animal Care (AAALAC). The surgical approach was similar to that described previously [18]. Adult male ferrets were acquired from an animal supplier (Marshall Farms). Head-post implantation surgeries were then performed in order to permit head-fixation during neurophysiology recordings. Two stainless steel head-posts were fixed to the animal along the midline using bone cement (Palacos), which bonded to the skull and to stainless steel screws that were inserted into the skull. After a two-week recovery period, animals were habituated to a head-fixed posture and auditory stimulation. At this point, a small (0.5–1 mm) craniotomy was opened above primary auditory cortex (A1) for neurophysiological recordings.

### Neurophysiology

Recording procedures followed those described previously [19, 20]. Briefly, upon opening a craniotomy, 1–4 tungsten micro-electrodes (FHC, 1–5 MΩ) were inserted to characterize the

tuning and response latency of the region of cortex. Sites were identified as A1 by characteristic short latency responses, frequency selectivity, and tonotopic gradients across multiple penetrations [21]. Subsequent penetrations were made with a 64-channel silicon electrode array [22]. Electrode contacts were spaced 20 $\mu$m horizontally and 25 $\mu$m vertically, collectively spanning 1.05 mm of cortex. Data were amplified (RHD 128-channel headstage, Intan Technologies), digitized at 30 KHz (Open Ephys [23]) and saved to disk for further analysis.

Spikes were sorted offline using Kilosort2 (https://github.com/MouseLand/Kilosort2). Spike sorting results were manually curated in phy (https://github.com/cortex-lab/phy). For all sorted and curated spike clusters, a contamination percentage was computed by measuring the cluster isolation in feature space. All sorted units with contamination percentage less than or equal to 5 percent were classified as single-unit activity. All other stable units that did not meet this isolation criterion were labeled as multi-unit activity. Both single and multi-units were included in all analyses.

## Acoustic stimuli

Digital acoustic signals were transformed to analog (National Instruments), amplified (Crown), and delivered through a free-field speaker (Manger) placed 80 cm from the animal's head and 30˚ contralateral to the the hemisphere in which neural activity was recorded. Stimulation was controlled using custom MATLAB software (https://bitbucket.org/lbhb/baphy), and all experiments took place inside a custom double-walled sound-isolating chamber (Professional Model, Gretch-Ken).

Auditory stimuli consisted of narrowband white noise stimuli with ≈0.3 octave bandwidth. In total, we presented fifteen distinct, non-overlapping noise bursts spanning a 5 octave range. Each noise was presented alone (-Inf dB) condition, or with a pure tone embedded at its center frequency for a range of different signal to noise ratios ($-10$dB, $-5$dB, $0$dB). Thus, each experiment consisted of 60 unique stimuli (4 SNR conditions X 15 center frequencies). Overall sound level was set to 60 dB SPL. Stimuli were 300ms in duration with 200ms ISI and each sound was repeated 50 times per experiment in a pseudo-random sequence.

## Bootstrapped estimates of decoding performance for different sample sizes

In Fig 6 panels d and g, we present the relative performance of *dDR* vs. *taPCA* and *stPCA* applied to real neural data across different sample sizes. Unlike our simulations, here we were restricted to a finite number of total trials ($k = 50$). Therefore, we utilized the following bootstrapping procedure to compute unbiased estimates of the standard error in cross-validated decoding performance at each sample size.

First, we selected a subset of the available trials ($k = 15$) to hold out for validation. Next, we re-sampled, with replacement, from the remaining data to build bootstrapped estimation sets. For example, for $k = 20$ this means that for each bootstrap sample we randomly selected 5 trials, with replacement, excluding the validation data. We then performed dimensionality reduction using *dDR*, *taPCA*, or *stPCA* and fit a decoding axis in the reduced dimensionality space for these 5 trials. Finally, we evaluated the decoding performance using this decoding axis on the held out $k = 25$ validation trials. We normalized the resulting decoding metric, $d'^2$, for *taPCA* and *stPCA* to the mean *dDR* $d'^2$ across all bootstraps for a given sample size. Thus, for each sample size $k$, we obtained a bootstrapped distribution of relative decoding performance between *dDR* and either *taPCA* or *stPCA*. The lines in Fig 6d, g represent the mean of this metric across bootstraps and the shading represents the standard deviation across bootstraps *i.e.*, the bootstrapped estimate of standard error [24]. For $k < 50$ we performed a bootstrap

correction on the standard error in order to account for re-sampling only a subset of our full $k$ = 50 trials sample [25].

## Results

### Neural population decoding and noise correlations

Decades of neurophysiology experiments have demonstrated that neural activity under most experimental conditions is variable. For example, in the auditory system, the number of action potentials a neuron produces in response to a sound stimulus varies each time that sound is presented. This stimulus-independent variability has often been attributed to stochastic noise. However, it is increasingly appreciated that latent physiological processes, such as changes in arousal and attention, could be driving these apparently spontaneous fluctuations in neural responses [15, 17, 26, 27].

Supporting the hypothesis that latent processes drive variability, modern neural recording techniques have demonstrated that stimulus-independent variability is often correlated between neurons. Thus, it cannot simply be attributed to independent noise in each neuron. Because this variability is not related to an experimentally controlled variable, like stimulus condition, it is commonly referred to as noise correlation. A rich literature exists describing the importance of noise correlation for understanding neural population codes (for a review, see [7]). For the purposes of our work, it is useful to briefly highlight some of the main concepts and define key terminology that we use throughout this manuscript. Readers familiar with previous work on pairwise noise correlation may wish to skip to the next section.

Noise correlation can be visualized by plotting the response distribution of a pair of neurons in state-space (Fig 1a–1c), where state-space refers to the euclidean space in which each axis represents the activity of a single neuron. If two neurons share a noise correlation, as is shown in Fig 1a–1c, their response distribution (illustrated by ellipses) will be elongated. This

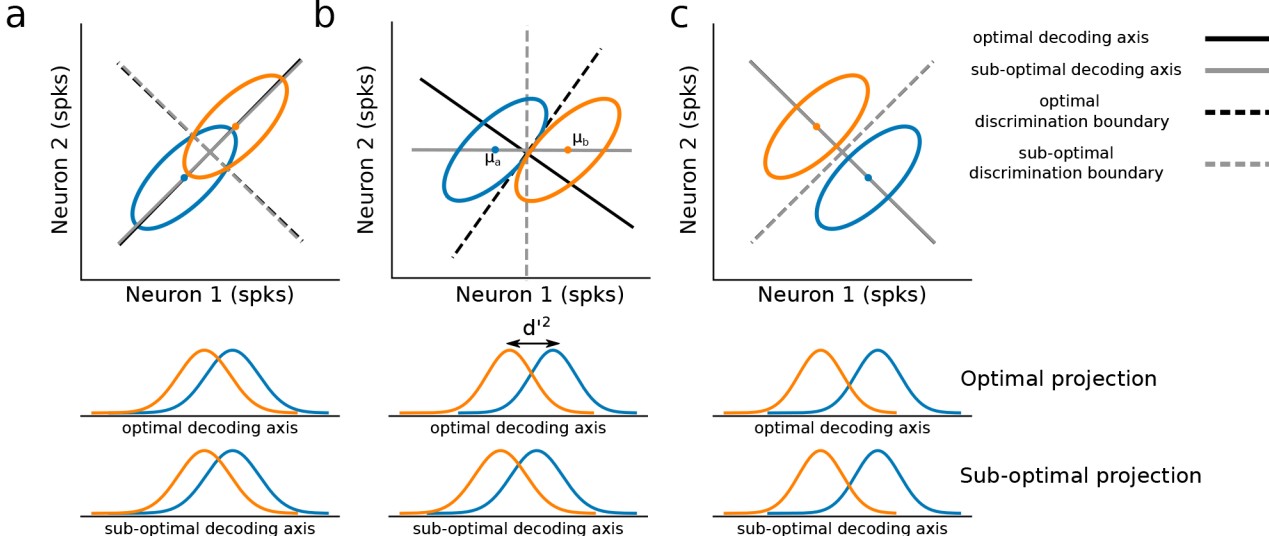

**Fig 1. Neuronal noise correlation and population coding. a.-c**. In each panel, the spiking responses of two neurons are simulated for two experimental conditions (blue vs. orange). Noise correlation strength and the absolute distance between the mean responses were fixed across all simulations. Top: Response distributions in state-space under each condition are summarized using an ellipse that shows one standard deviation of responses around the mean. Middle: Responses are projected onto the optimal linear decoding axis, where decoding metrics such as $d'^2$ can be visualized and measured. $d'^2$ quantifies how discriminable two Gaussian distributions are. Bottom: Responses are projected onto a sub-optimal decoding axis. Unlike the optimal axis, this does not take into account noise correlations. Figure is adapted from Averbeck & Lee, 2006 [6].

reflects the observation that when one neuron fires more spikes than average, the other neuron tends to do so as well. The sign and strength of a noise correlation is reflected by the shape of this response distribution.

Theoretical work has shown that the ability of a population of neurons to discriminate between different stimulus conditions critically depends on how noise correlation interacts with sensory tuning. To illustrate this point, Fig 1a–1c shows simulated response distributions for two neurons under two different stimulus conditions (blue vs. orange). When the noise correlation is aligned with the coding axis, it interferes with the ability to discriminate between the two distributions (Fig 1a). However, when the noise correlation is orthogonal, discrimination is actually easier (Fig 1c). This makes sense intuitively—if the uncontrolled variability (noise correlation) changes neural activity in the same way as the stimulus, then it is impossible to know if a change in the activity should be attributed to stimulus, or to noise.

In practice, noise correlation need not be perfectly aligned with, or orthogonal to, the coding axis. The specific alignment of noise correlation can be leveraged to achieve an optimal decoding strategy. The intuition for this optimization is illustrated in Fig 1b. The linear decoding axis is rotated to minimize the amount of noise correlation observed by a *e.g.* downstream readout neuron (Fig 1c, middle) relative to a sub-optimal decoding strategy that only takes into account the trial-averaged activity of each response distribution (Fig 1b, bottom). Thus, accurately measuring noise correlation is important for optimally decoding neural population activity.

## Small sample sizes limit the reliability of neural decoding analysis

Linear decoding identifies a linear, weighted combination of neural activity along which distinct experimental conditions (*e.g.* different sensory stimuli) can be discriminated. In neural state-space, this weighted combination is referred to as the decoding axis, $w_{opt}$, the line along which the distance between stimulus classes is maximized and trial-trial variance is minimized (Fig 2a and 2b). To quantify decoding accuracy, single-trial neural activity is projected onto this axis and a decoding metric is calculated to quantify the discriminability of the two stimulus classes. Here, we use $d'^2$, the discrete analog of Fisher Information [4, 5]. This discriminability metric has been used in a number of previous studies [6, 9–11, 28] and has a direct relationship to classical signal detection theory [4, 29].

Looking at the simulated data in Fig 2a and 2b, one can appreciate that an accurate estimate of $w_{opt}$ requires knowledge of both the mean response evoked by each stimulus class ($\mu_a$ vs. $\mu_b$), as well the population noise correlations, $\Sigma$ (summarized by the ellipses in Fig 2a and 2b). Indeed, $d'^2$, is directly dependent on these features:

$$d'^2 = \Delta\mu^T w_{opt} \tag{1}$$

$$w_{opt} = \Sigma^{-1}\Delta\mu \tag{2}$$

$$\Delta\mu = \mu_a - \mu_b \tag{3}$$

Where $\mu_a$ and $\mu_b$ are the $N$x1 vectors describing the mean response of an $N$-neuron population to two stimuli, *a* vs. *b*, respectively, and $\Sigma$ is the average $N$ x $N$ covariance matrix $\frac{1}{2}(\Sigma_a + \Sigma_b)$ (*e.g.* Fig 2c).

In practice, the noise correlations between neurons (or $r_{sc}$) is reported to be very small—on the order of $10^{-1}$ or $10^{-2}$ [30–32]. As we can see from the shuffled distribution in Fig 2a (bottom), this can pose a problem for accurate estimates of the off-diagonal elements in $\Sigma$, and, as a consequence, $w_{opt}$ itself. This difficulty is especially pronounced when sample sizes are

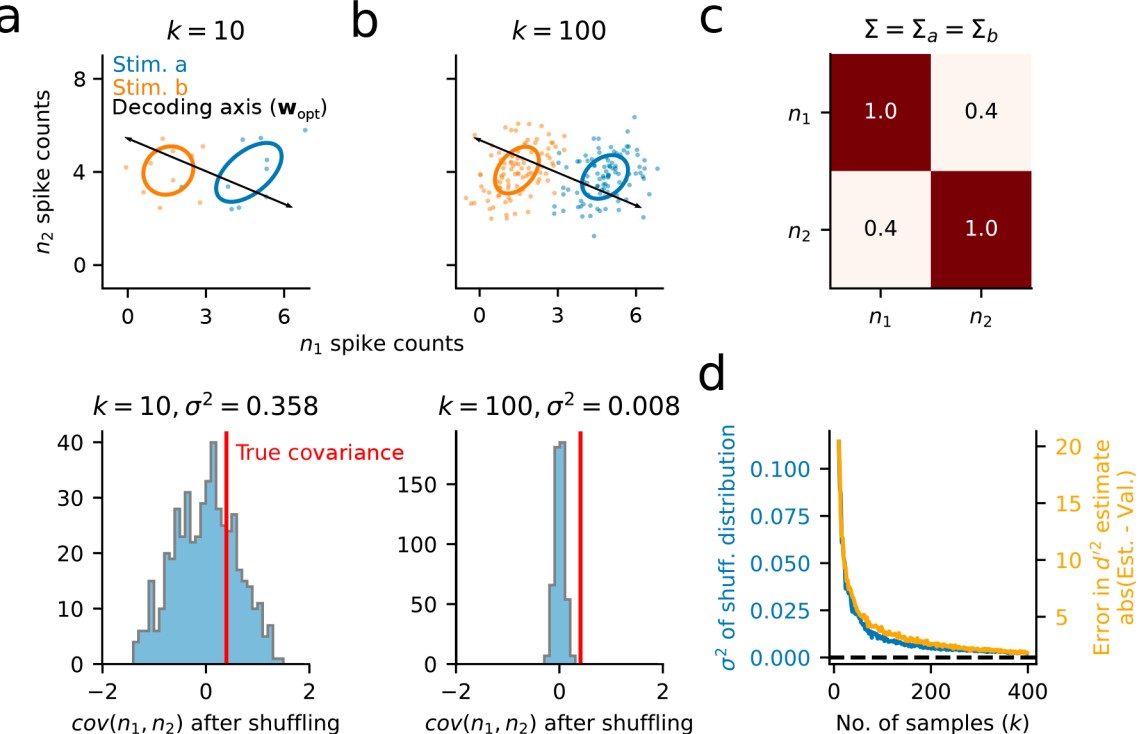

**Fig 2. Measurements of noise correlations and discriminability are unreliable when sampling is limited. a**. Top: $k = 10$ single trial spike count responses are drawn from standard multivariate Gaussians $\mathcal{N}(\mu_a, \Sigma)$ and $\mathcal{N}(\mu_b, \Sigma)$ corresponding to two different stimulus conditions, $a$ and $b$. Ellipses show the standard deviation of spike counts across trials. Bottom: Reliability of the noise correlation estimate between neuron 1 ($n_1$) and neuron 2 ($n_2$) is calculated by shuffling values of $n_1$ 500 times. The true covariance (red line) falls within this distribution, indicating that estimates of covariance are not reliable for $k = 10$. **b**. Same as in (a), but drawing $k = 100$ samples for each stimulus. The narrower distribution of permuted measures indicates a greater likelihood of identifying an accurate estimate of covariance. **c**. The covariance matrix, $\Sigma$, used to generate data in (a)/(b). The true pairwise covariance for this pair of simulated neurons has a value of 0.4. **d**. Variance ($\sigma^2$) of covariance estimates based on the permutation analysis in (a)/(b) for a range of sample sizes, $k$ (blue). Variance decays as $\mathcal{O}(\frac{1}{k-1})$ (see S1 Appendix). Overlaid is the difference in stimulus discriminability, $d'^2$ (Eq 1), between estimation and validation sets (50–50 split) estimated for each sample size (orange). Large values in the $d'^2$ difference for low $k$ indicate overfitting of $w_{opt}$ to the estimation data. This difference asymptotes toward zero as sample size increases and the estimate of covariance becomes reliable.

relatively small (compare Fig 2a to 2b). The estimates of covariance and stimulus discriminability improve with increasing sample size, but robust performance is not reached until ≈100 stimulus repetitions, even for this case with relatively strong covariance (Fig 2d). The sample sizes (*e.g.* number of trials) in most experiments, especially those involving animal behavior, are typically much lower, raising the question: How can one reliably quantify coding accuracy in large neural populations observed over relatively few trials?

## Neural population activity is low-dimensional

Analysis of neural population data with dimensionality reduction has consistently revealed low-dimensional structure in neural activity [33]. Specifically, recent studies have found that this noise correlation is dominated by a small number of latent dimensions [14, 15, 17, 27]. Noise correlation impacts stimulus coding accuracy [7] and is known to depend on internal states, such as attention, that affect behavioral task performance [15, 16, 30, 34, 35]. These findings suggest that the space of neural activity relevant for understanding stimulus decoding,

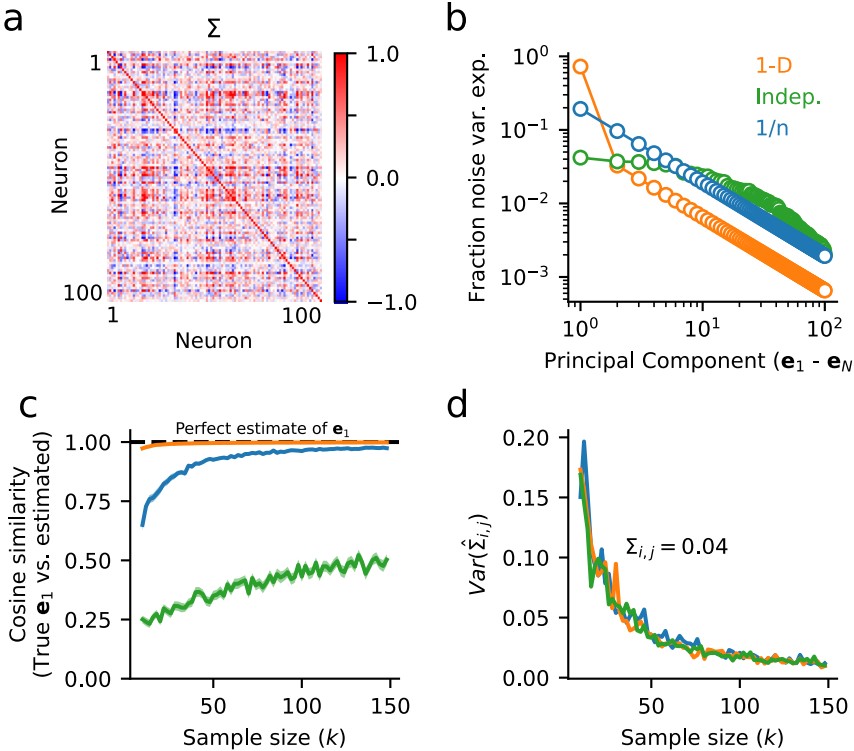

**Fig 3. Low-dimensional noise correlation can be estimated reliably for neural populations, even when pairwise noise correlation cannot. a**. Example covariance matrix, $\Sigma$, for a 100-neuron population with low-dimensional covariance structure. **b**. Scree plot shows the fraction of total population variance captured along each noise dimension, computed by *PCA*, for three different datasets with varying dimensionality. Orange: 1-dimensional noise (1-D), covariance matrix in (a); green: independent noise (Indep.); blue: power law decay ($1/n$). **c**. Surrogate datasets with varying numbers of samples, $k$, are drawn from the three noise distributions in (b). For each dataset, the cosine similarity between the estimate of the largest noise dimension, $\hat{e}_1$, and the true noise dimension, $e_1$, is plotted as function of sample size. For low-dimensional data, $e_1$ can be estimated very reliably. **d**. Variance in the estimate of covariance, $\Sigma_{i,j}$, for two neurons with a true covariance of 0.04 is plotted as a function of the number of trials, as in Fig 2d. Even at sample sizes >100, $Var(\hat{\Sigma}_{i,j}) \approx 0.02$, corresponding to a standard deviation of $\approx 0.14$. Therefore, estimates of $\Sigma_{i,j}$, may be off by up to an order of magnitude. Note that the amount of uncertainty does not depend on the dimensionality of the data, and results for all three datasets overlap (see S1 Appendix for an analytical derivation).

and its relationship to behavior, may be small relative to the total number of recorded neurons.

When population data exhibits low-dimensional structure, the largest eigenvectors of $\Sigma$ (*i.e.* the top principal components of population activity) provide a reasonable, low-rank approximation to the full-rank covariance matrix. Importantly, these high variance dimensions of covariability can be estimated accurately even from limited samples. To illustrate this point, we simulated population spike counts, $X$, for $N = 100$ neurons by drawing $k$ samples from a multivariate Gaussian distribution with mean $\mu$ and covariance $\Sigma$ (Eq 4).

$$X = \mathcal{N}(\mu, \Sigma) + \epsilon_{indep.} \tag{4}$$

Where in Eq 4, $\epsilon_{indep.}$ represents a small amount of independent noise added to each neuron, effectively removing any significant structure in the smaller noise modes.

To investigate how different noise structures impact estimates of $\Sigma$, we simulated three different surrogate populations. First, we simulated data with just one large, significant noise dimension (Fig 3b–3d, 1-D data, orange). In this case, the first eigenvector can be estimated

reliably, even from just a few samples (Fig 3c). However, when the noise is independent and shared approximately equally across all neurons, estimates of the first eigenvector are poor (Fig 3c, Indep. noise, green). These first two simulations represent extreme examples—in practice, population covariability tends to be spread across at least a few significant dimensions [36]. To investigate a scenario that more closely mirrors this structure, we simulated a third dataset where the noise eigenspectrum decayed as $1/n$, where $n$ goes from $n = 1$ to $N$. Recent studies of large neural populations suggest that this power law relationship is a reasonable approximation to real neural data [36]. In this case, by $k \approx 50$ trials, estimates of the first eigenvector are highly reliable, approaching a cosine similarity of $\approx 0.9$ between the estimated and true eigenvectors (Fig 3c, $1/n$ noise, blue). In all simulations, regardless of dimensionality, we find that estimates of single elements of $\Sigma$ (*i.e.* single noise correlation coefficients) are highly unreliable (Fig 3d), as we see in the two-neuron example (Fig 2d).

Collectively, these simulations demonstrate that accurate estimates of noise correlation need not necessarily be limited by uncertainty in estimates of individual noise correlation coefficients themselves. In the following sections we describe a simple decoding-based dimensionality reduction algorithm, *dDR*, that leverages low-dimensional structure in neural population activity to facilitate reliable measurements of neural decoding.

## Decoding-based Dimensionality Reduction (*dDR*)

The *dDR* algorithm operates on a pairwise basis. That is, given a set of neural data collected over $S$ different conditions, a different *dDR* projection exists for each of the $\frac{S!}{2!(S-2)!}$ unique pairs. For simplicity, we will describe the case where $S = 2$, and consider these to be two unique stimulus conditions. However, note that the method can be applied in exactly the same manner to handle datasets with many different types and numbers of decoding conditions, where a unique *dDR* projection would then exist for each pair.

Let us consider the spiking response of an $N$-neuron population evoked by two different stimuli, $S_a$ and $S_b$, over $k$-repetitions of each stimulus. From this data we form two response matrices, $A$ and $B$, each with shape $N$ x $k$. Remembering that our goal is to estimate discriminability ($d'^2$, Eq 1), the *dDR* projection should seek to preserve information about both the mean response evoked by each stimulus condition, $\mu_a$ and $\mu_b$, as well as the noise correlations, $\Sigma$. Therefore, we define the first dimension of *dDR* to be the axis that maximally separates $\mu_a$ and $\mu_b$. We call this the *signal* axis.

$$signal = \mu_a - \mu_b = \Delta\mu \tag{5}$$

Next, we compute the first eigenvector of $\Sigma$, $e_1$. This represents the largest noise mode of the neural population activity. Together, *signal* ($\Delta\mu$) and $e_1$ span the plane in state-space that is most optimized for reliable decoding. Finally, to form an orthonormal basis, we define the second *dDR* dimension as the axis orthogonal to $\Delta\mu$ in this plane. As this second dimension is designed to preserve noise covariance, we call this the $noise_1$ axis.

$$noise_1 = e_1 - e_1\Delta\mu^T \tag{6}$$

The process outlined above is schematized graphically in Fig 4.

Thus, the *signal* and $noise_1$ axes make up a 2 x $N$ set of weights, analogous to the loading vectors in standard *PCA*, for example. By projecting our $N$ x $k$ data onto this new basis, we capture both the stimulus coding dimension ($\Delta\mu$) and preserve the principal noise correlation dimension ($e_1$), two critical features for measuring stimulus discriminability. Importantly, because $e_1$ can be measured more robustly than $\Sigma$ itself (Fig 3), performing this dimensionality

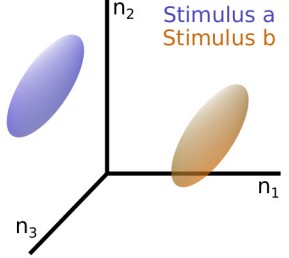
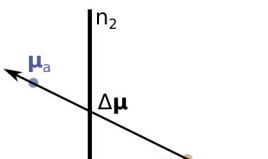
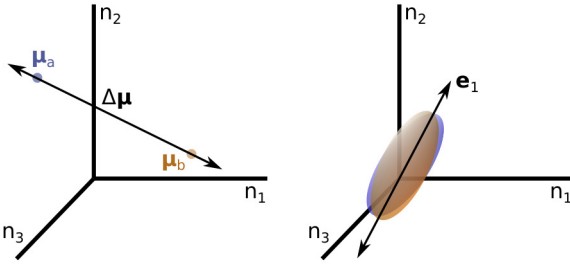
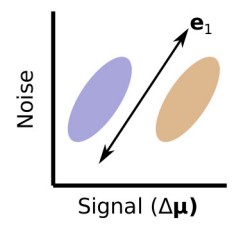

**Fig 4. Decoding-based Dimensionality Reduction (dDR).** Left to right: Responses of 3 neurons ($n_1$, $n_2$, $n_3$) to two different stimuli are schematized in state-space. Ellipsoids illustrate the variability of responses across trials. **1**. To perform *dDR*, first the difference is computed between the two mean stimulus responses, $\Delta\mu$. **2**. Next, the mean response is subtracted for each stimulus to center the data around 0, and *PCA* is used to identify the first eigenvector of the noise covariance matrix, $e_1$ (additional noise dimensions $e_m$, $m > 1$ can be computed, see text). **3**. Finally, the raw data are projected onto the plane defined by $\Delta\mu$ and $e_1$.

reduction helps mitigate the issues we encounter due to small sample sizes and large neural datasets.

As mentioned in the previous section, neural data often contains more than one significant dimension of noise correlation. To account for this, *dDR* can easily be extended to include more noise dimensions. To include additional dimensions, we deflate the spike count matrix, $X$, by subtracting out the *signal* and *noise*$_1$ dimensions identified by standard *dDR*, then perform *PCA* on the residual matrix to identify $m$ further *noise* dimensions. Note, however, that for increasing $m$ the variance captured by each dimension gets progressively smaller. Therefore, estimation of these subsequent noise dimensions becomes less reliable and will eventually become prone to over-fitting, especially with small sample sizes. For this reason, care should be taken when extending *dDR* in this way.

To demonstrate the performance of the *dDR* method, we generated three sample datasets containing $N = 100$ neurons and $S = 2$ stimulus conditions. Each of the three datasets contained unique noise correlation structure: 1. $\Sigma$ contained one significant dimension (Fig 5a) 2. $\Sigma$ contained two significant dimensions (Fig 5b) 3. Noise correlation decayed as $1/n$ (Fig 5c). For each dataset, we measured cross-validated $d'^2$ between stimulus condition $a$ and stimulus condition $b$ using standard *dDR* with one noise dimension ($dDR_1$), with two noise dimensions ($dDR_2$), or with three noise dimensions ($dDR_3$). We also estimated $d'^2$ using the full-rank data, without performing *dDR*. The bottom panels of Fig 5a–5c plot the decoding performance of each method as a function of sample size (*i.e.* number of stimulus repetitions). In each case, $d'^2$ is normalized to the asymptotic performance of the full-rank approach, when the number of samples is $>>$ than the number of neurons. This provides an approximate estimate of true discriminability for the population.

In contrast to the full-rank data where overfitting leads to dramatic underestimation of $d'^2$ on the test data for most sample sizes (Fig 5a–5c, bottom, grey lines), we find that $d'^2$ estimates after performing *dDR* are substantially more accurate and, critically, more reliable across sample sizes. That is, asymptotic performance of the *dDR* method is reached much more quickly than for the full-rank method.

For the one-dimensional noise case, note that there is no benefit of including additional *dDR* dimensions (Fig 5a), while for the higher dimensional data shown in Fig 5b and 5c, we see some improvements with $dDR_2$ and $dDR_3$. However, these benefits don't begin to appear

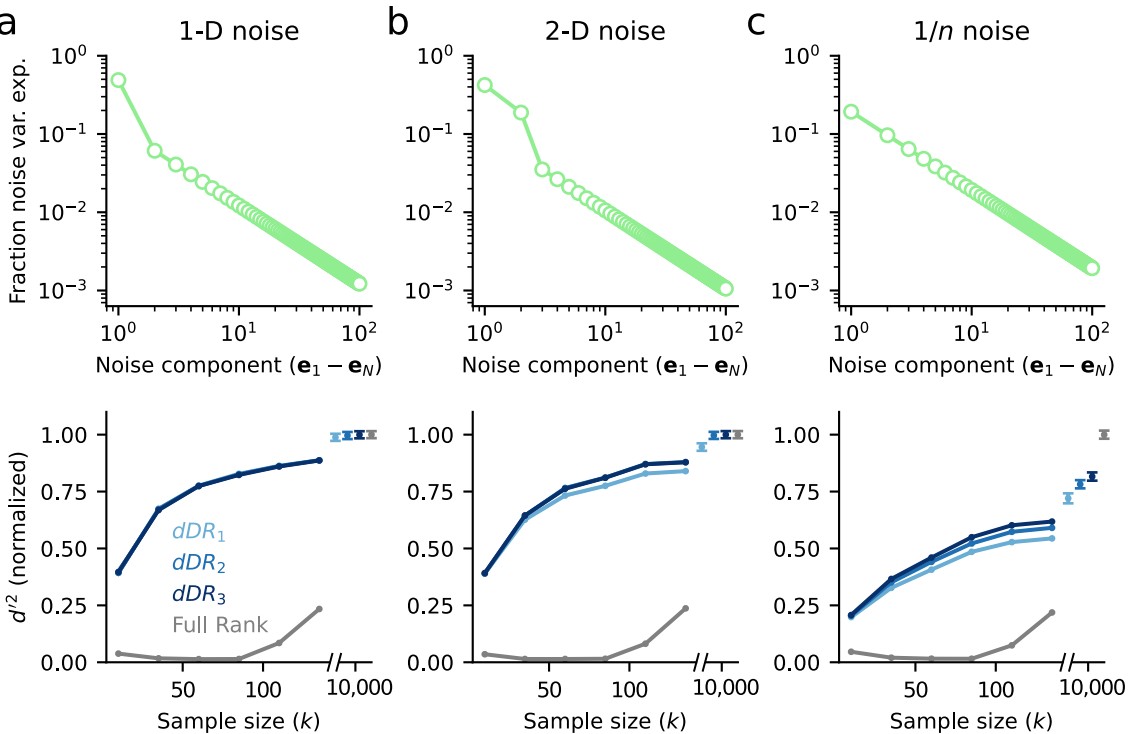

**Fig 5. Evaluation of decoding accuracy and reliability with *dDR*. a**. Analysis of data with one-dimensional (1-D) noise covariance. For each sample size, $k$, 100 datasets were generated from the same multivariate Gaussian distribution (Eq 4) where Σ was a rank-one covariance matrix and the mean response vector, $\mu$, corresponded to one of two stimulus conditions, $a$ or $b$. Top: Scree plot of noise covariance. Bottom: Cross-validated discriminability, $d'^2$, between $a$ and $b$ computed with full-rank data and with *dDR* using one ($dDR_1$), two ($dDR_2$) or three ($dDR_3$) noise dimensions, as a function of sample size. Mean $d'^2$ across all 100 surrogate datasets is shown here. For $k >> N$, the *dDR* results converge to the asymptotic value of the full-rank $d'^2$. However, even for small $k$, the *dDR* analyses estimates are much more accurate than the full-rank approach. **b**. Same as in (a), but for two-dimensional noise covariance data. In this case, $dDR_2$ captures the second noise dimension and outperforms the standard 1-D approach ($dDR_1$) **c**. Same as in (a) and (b), but for $1/n$ noise covariance.

until $k$ becomes large and they diminish with increasing noise dimensions—the improvement of $dDR_2$ over $dDR_1$ is larger than that of $dDR_3$ to $dDR_2$ Fig 5b and 5c. This is because subsequent noise dimensions are, by definition, lower variance and therefore more difficult to estimate reliably from limited sample sizes.

## *dDR* recovers more decoding information than standard principal component analysis

One popular method for dimensionality reduction of neural data is principal component analysis (*PCA*) [33]. Generally speaking, *PCA* can be implemented on neural data in one of two ways: application to single trial spike counts *PCA* or trial-averaged spike counts *PCA*. In the single trial approach (*stPCA*), principal components are measured across all single trials and all experimental conditions. The resulting *PCs* capture variance both across trials and across e. g. stimulus conditions. In trial-averaged *PCA* (*taPCA*), single trial responses are first averaged per experimental condition and *PCs* are measured over the resulting $N$-neuron x $S$-condition spike count matrix. In this case, for different stimulus conditions, the *PCs* specifically capture variance of stimulus-evoked activity rather than trial-trial variability, making it a more logical choice for many decoding applications. In the case of $S = 2$, as we have outlined above for the *dDR* illustration (Fig 4), *taPCA* is equivalent to $\Delta\mu$, the first *dDR* dimension. Thus, *dDR* can

roughly be thought of as a way to combine *taPCA* and *stPCA*—*taPCA* identifies the *signal* dimension and *stPCA* identifies the *noise* dimension(s).

To demonstrate the relative decoding performance achieved using each method, we applied each to a dataset collected from primary auditory cortex in an awake, passively listening ferret. $N = 52$ neurons were recorded simultaneously using a 64-channel laminar probe [22] as in [19, 20, 37]. Auditory stimuli consisting of narrowband (0.3 octave bandwidth) noise bursts were presented alone (-Inf dB) or with a pure tone embedded at varying SNRs (0 dB, −5 dB, −10 dB) in the hemifield contralateral to the recording site (see Experimental Methods). Each stimulus was repeated 50 times. The neural response to each stimulus was defined as the total number of spikes detected during the 300 ms stimulus duration for each neuron. For *stPCA* and *dDR*, we selected only the top $m = 2$ total dimensions, and for *taPCA*, we selected the single dimension, $\Delta\mu$, that exists for $S = 2$. This dataset allowed us to investigate how each dimensionality reduction method performs for two distinct, behaviorally relevant neural decoding questions: One, how well can neural activity perform fine discriminations (*tone-in-noise detection*), discriminating noise alone vs. noise with tone? Two, how well can it perform coarse discriminations (*frequency discrimination*), discriminating noise centered at frequency A vs. noise at frequency B?

The A1 dataset displayed a range of frequency tuning (Fig 6a), with the majority of units tuned to ≈3.5 kHz. We therefore defined this as the best frequency of the recording site (on-BF, Fig 6b). For *tone detection*, we measured discriminability ($d'^2$, Eq 1) between on-BF noise alone (on-BF, -Inf dB) and on-BF noise plus tone (on-BF, −5 dB), which each drove similar sensory responses (Fig 6b and 6c). For *frequency discrimination*, we measured discriminability between the neural responses to on-BF noise and off-BF noise, where off-BF was defined as ≈1 octave away from BF, and drove a very different population response (Fig 6b and 6f). In both cases, *taPCA* and *dDR* outperformed *stPCA* (Fig 6d and 6g). This first result is unsurprising due to the fact that *stPCA* is the only method not explicitly designed to capture variability in the sensory response. The top *PCs* are dominated by dimensions of trial-trial variability that do not necessarily contain stimulus information and thus underestimate $d'^2$ relative to the other two methods.

We also find that *dDR* consistently performs as well or better than *taPCA*. For the *tone detection* data, the sensory signal ($\Delta\mu$) is small (*i.e.*, trial-averaged responses to the two stimuli were similar) and covariability is partly aligned with $\Delta\mu$. Under these conditions, *dDR* makes use of correlated activity to optimize the decoding axis ($w_{opt}$) and improve discriminability. *taPCA*, on the other hand, has no information about these correlations and is therefore equivalent to projecting the single trial responses onto the *signal* axis, $\Delta\mu$. Thus, it underestimates $d'^2$ (Fig 6c and 6d). In the *frequency discrimination* example, $\Delta\mu$ is large. The covariability has similar magnitude to the previous example, but it is not aligned to the discrimination axis, and thus has no impact on $w_{opt}$. In this case, *dDR* and *taPCA* perform similarly (Fig 6f and 6g). These examples highlight that under behaviorally relevant conditions, *dDR* can offer a significant improvement over standard *PCA*, even with as few as 20 trials.

## Discussion

We have described a simple new method for robust decoding analysis of neural population data, decoding-based dimensionality reduction (*dDR*). This approach combines strategies for both trial-averaged *PCA* and single-trial *PCA* to identify important dimensions of population activity that govern neural coding accuracy. Using both simulated and real neural data, we demonstrated that the method performs robustly for neural decoding analysis in low experimental trial count regimes where the performance of full-rank methods break down. Across a

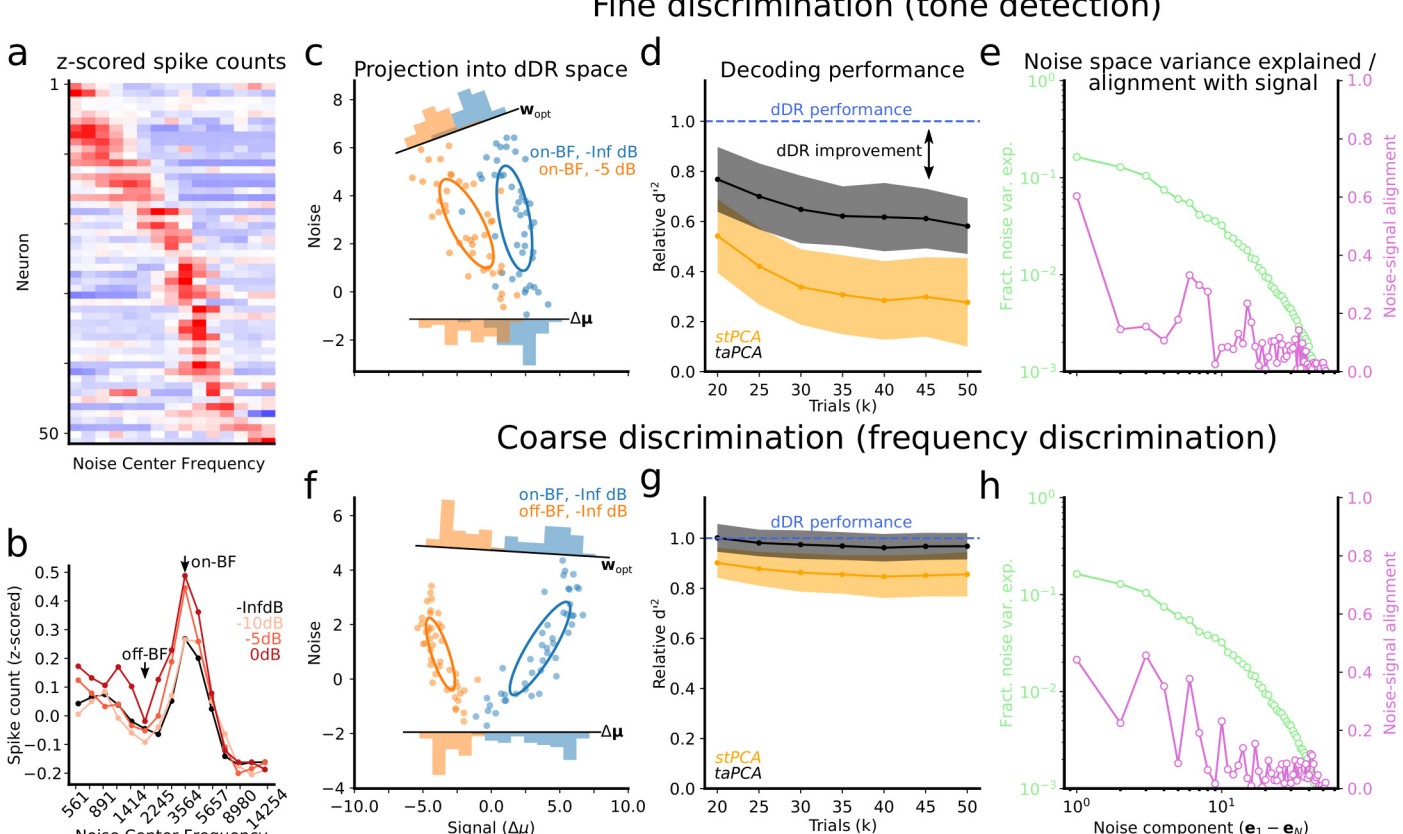

**Fig 6. *dDR* outperforms *PCA* for fine sensory discrimination. a**. Heatmap shows mean *z*-scored spike counts of *N* = 52 simultaneously recorded units for 15 different narrowband noise bursts (0.3 octave bandwidth tiling 5 octaves, *x*-axis). Each row shows tuning for one unit, with red indicating higher firing rate response. *x*-axis (Noise Center Frequency of the sound stimulus) is shared with panel b. **b**. Population tuning curve for noise alone (black, data from panel a) and noise plus −10, −5, and 0 dB tones (light to dark red), computed by averaging tuning curves across neurons. **c-e**. Decoding analysis for tone-in-noise detection. **c**. Scatter plot compares single trial responses to noise alone at best frequency (on-BF, blue) vs. noise + −5dB tone (orange), projected into *dDR* space. Ellipses show standard deviation across trials, marginal histograms show projection of data onto optimal decoding axis ($w_{opt}$) or onto Δμ (equivalent to performing trial-averaged *PCA*). **d**. Estimate of relative $d'^2$ as a function of sample size (number of trials, *k*) using each dimensionality reduction method. For each data point, relative $d'^2$ was measured between *taPCA* vs. *dDR* and *stPCA* vs. *dDR*. This metric was averaged over 200 bootstrap samples of *k* trials. Shading indicates standard error. See Methods for details. **e**. Fraction variance explained by each noise component (green) computed by performing *PCA* on mean-centered single trial data. The alignment of each noise component with the signal axis is shown in purple. **f-h** Same as panels (c)-(e), for noise alone on-BF vs. noise alone off-BF (see panel b).

range of behaviorally relevant stimulus conditions, *dDR* consistently performs as well or better than standard principal component analysis.

## Applications

*dDR* is designed to optimize the performance of linear decoding methods in situations where sample sizes are small. This is often the case for neurophysiology data collected from behaving animals, where the number of stimulus and/or behavior conditions are fundamentally limited by task performance. In these situations, using full-rank decoding methods is unfeasible as it leads to dramatic overfitting and unreliable performance [12]. Dimensionality reduction methods, such as *PCA*, can be used to mitigate overfitting issues. However, the correct implementation of *PCA* in neural data is often ambiguous, and multiple different approaches to dimensionality reduction have been proposed [33]. We suggest *dDR* as a simple, standardized alternative that captures the strengths of different *PCA* approaches. Unlike conventional *PCA*,

the *signal* and *noise* axes that comprise the *dDR* space have clear interpretations with respect to neural decoding. Importantly, *dDR* components explicitly preserve stimulus-independent population covariability. In addition to being important for overall information coding, this covariability is known to depend on behavior state [15, 16, 30, 34, 38] and stimulus condition [31, 39–41]. Therefore, approaches that do not preserve these dynamics, such as trial-averaged *PCA*, may not accurately characterize how information coding changes across varying behavior and/or stimulus conditions.

## Interpretability and visualization

A key benefit of *dDR* is that the axes making up the *dDR* subspace are easily interpretable: The first axis (*signal*) represents the dimension with maximal information about the difference in evoked activity between the two conditions to be decoded, and the second (*noise*) axis captures the largest mode of condition-independent population covariability in the data. Therefore, within the *dDR* framework it is straightforward to investigate how this covariability interacts with discrimination, an important question for neural information coding. Further, standard *dDR* (with a single noise dimension) can be used to easily visualize high-dimensional population data, as in Fig 6. For methods like *PCA*, it can be difficult to dissociate signal and noise dimensions, as the individual principal components can represent an ambiguous mix of task conditions, stimulus conditions, and trial-trial variability [42]. Moreover, with *PCA* the number of total dimensions is typically selected based on their cumulative variance explained, rather than by selecting the dimensions that are of interest for decoding, as in *dDR*.

## Extensions

**Latent variable estimation.** *dDR* makes the assumption that latent sources of low-dimensional neural variability can be captured using simple, linear methods, such as *PCA*. While these methods often seem to recover meaningful dimensions of neural variability [16], a growing body of work is investigating new, alternative methods for estimating these latent dynamics [15, 17, 43–45], and this work will continue to lead to important insights about the nature of shared variability in neural populations.

We suggest that *dDR* can be extended to incorporate these new methods. For example, rather than defining *dDR* on a strictly per decoding pair basis, a global noise axis could be identified across all experimental conditions using a custom latent variable method. This could then be applied to the decoding-based dimensionality reduction such that the resulting *dDR* space explicitly preserves activity in this latent space to investigate how it interacts with coding.

**Incorporating additional *dDR* dimensions.** In this work we have described *dDR* primarily as a transformation from *N*-dimensions to two dimensions, *signal* and *noise*, with the exception of Fig 5. In our code repository, https://github.com/crheller/dDR, we include examples that demonstrate how the *dDR* method can be extended to include additional dimensions. However, as discussed in the main text, it is important to remember that estimates of neural variability beyond the first principal component may become unreliable as variance along these dimensions gets progressively smaller, especially in low trial regimes. In short, while information may be contained in dimensions $>m = 2$, caution should be used to ensure that these dimensions can be estimated reliably.

## Related methods for dimensionality reduction and neural decoding

A growing number of techniques exist for performing dimensionality reduction on neural data [33]. It is outside the scope of this work to provide an exhaustive review of these methods, however, it is useful to highlight a couple of methods that share similarities with *dDR*.

Standard principal component analysis (*PCA*) remains the most commonly applied method for dimensionality reduction of neural data. Indeed, many other methods, (*e.g.* k-means clustering, non-negative matrix factorization etc.), can simply be viewed specially constrained versions of *PCA* [46]. In the case of *dDR*, the key distinction from *PCA* is that *dDR* explicitly preserves information about both trial-trial neural variability and mean, *e.g.* stimulus evoked, activity. *PCA*, on the other hand, is an entirely unsupervised method. Therefore, individual principal components often contain a mixture of trial-trial variability and mean activity, making their interpretation challenging. Furthermore, performing *PCA* can lead to sub-optimal neural decoding, as we demonstrate above.

Recently, Kobak et al. developed a powerful method called demixed *PCA* (*dPCA*) which produces interpretable low-dimensional representations of neural population data [42]. While this work shares some conceptual similarities with *dDR*, namely that both measure interpretable, low-dimensional representations of neural data, their applications are distinct. With *dPCA*, the idea is to produce components that allow accurate decoding of *e.g.* stimulus condition and also maintain a faithful representation of the underlying neural state space geometry. In the context of optimal decoding, these two aims can sometimes be at odds—the best geometrical representation of the data does not necessarily lead to optimal decodability. This trade off is illustrated nicely in their manuscript [42].

Unlike *dPCA*, *dDR*, only seeks to maximize information that can be used for decoding. Therefore, *dDR* can be thought of as a preprocessing step that should be applied prior to performing standard decoding methods, such as linear discriminant analysis (*LDA*). Further, *dDR* is designed for optimal decoding of only two experimental conditions at a time. *dPCA* is not restricted in this pairwise way. Therefore, *dPCA* is useful when an interpretable, constant low-*D* space across many different experimental conditions is desired, while *dDR* should be used when optimal decoding is the goal.

## Code availability

We provide Python code for *dDR* which can be downloaded and installed by following the instructions at https://github.com/crheller/dDR. We also include a short demo notebook that highlights the basic work flow and implementation of the method to simulated data. All code used to generate the figures in this manuscript is available in the repository.

## Supporting information

**S1 Appendix.**
(PDF)

## Author Contributions

**Conceptualization:** Charles R. Heller, Stephen V. David.

**Data curation:** Charles R. Heller.

**Formal analysis:** Charles R. Heller.

**Funding acquisition:** Charles R. Heller, Stephen V. David.

**Investigation:** Charles R. Heller.

**Methodology:** Charles R. Heller.

**Project administration:** Stephen V. David.

**Resources:** Stephen V. David.

**Software:** Charles R. Heller.

**Supervision:** Stephen V. David.

**Validation:** Stephen V. David.

**Visualization:** Charles R. Heller.

**Writing – original draft:** Charles R. Heller.

**Writing – review & editing:** Charles R. Heller, Stephen V. David.

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
