## [Decision Letter · Decision Letter 0]

19 Apr 2022

PONE-D-22-02203Targeted dimensionality reduction enables reliable estimation of neural population coding accuracy from trial-limited dataPLOS ONE

Dear Dr. Heller,

Thank you for submitting your manuscript to PLOS ONE. After careful consideration, we feel that it has merit but does not fully meet PLOS ONE’s publication criteria as it currently stands. Therefore, we invite you to submit a revised version of the manuscript that addresses the points raised during the review process.

This manuscript describes a powerful new technique for the analysis of neural data that has potential applications in sensory physiology. However, the presentation could be improved by a clearer description of the methods. Furthermore, the use of additional principal components (not just the first) should be explored and its possible advantages discussed, as requested by Reviewer #1. Please address the other comments by the Reviewers.

We look forward to receiving your revised manuscript.

Kind regards,

David S Vicario, Ph.D.

Academic Editor

PLOS ONE

Journal Requirements:

2. Please amend either the title on the online submission form (via Edit Submission) or the title in the manuscript so that they are identical.

“This work was supported by a National Science Foundation Graduate Research Fellowship (NSF GRFP, GVPRS0015A2) (CRH), the National Institute of Health (NIH, R01 DC0495) (SVD), Achievement Rewards for College Scientists (ARCS) Portland chapter (CRH), and by the Tartar Trust at Oregon Health and Science University (CRH).”

“This work was supported by a National Science Foundation Graduate Research Fellowship (NSF GRFP, GVPRS0015A2) (CRH), the National Institute of Health (NIH, R01 DC0495) (SVD), Achievement Rewards for College Scientists (ARCS) Portland chapter (CRH), and by the Tartar Trust at Oregon Health and Science University (CRH). The funders had no role in study design, data collection and analysis, decision to publish, or preparation of the manuscript.”

Reviewers' comments:

Reviewer's Responses to Questions

**Comments to the Author**

1. Is the manuscript technically sound, and do the data support the conclusions?

Reviewer #1: Yes

Reviewer #2: Yes

2. Has the statistical analysis been performed appropriately and rigorously? 

Reviewer #1: Yes

Reviewer #2: Yes

3. Have the authors made all data underlying the findings in their manuscript fully available?

Reviewer #1: Yes

Reviewer #2: Yes

4. Is the manuscript presented in an intelligible fashion and written in standard English?

Reviewer #1: No

Reviewer #2: Yes

5. Review Comments to the Author

Reviewer #1: In the current work, the authors proposed a novel computation approach to decode high-dimensional neural data. In this approach, the authors proposed to use the mean of different experimental conditions and the first eigenvector of the population covariance to determine the decoding axis, which has the potential to achieve better decoding performance with limited trials than the traditional PCA. The work is technically solid and interesting.

A major issue is the very abbreviated description of the rationale and technical details. For example, the first section of Results described some very basic concepts of this approach. But for anyone who has not read Averbeck and Lee 2006, it would be very hard to understand.

Other major concerns:

1. To show the advantage of the new method over traditional PCA, authors applied each method to neural data collected from auditory cortices. The general result did show a clear advantage of the new method over PCA. However, in this comparison, only the first two PCs were used to decode stimulus conditions that have two dimensions. In analyses of auditory neural data, the first PC of PCA usually reflects the rise and fall of the sound, while the second and third PCs contain information about stimulus conditions. Therefore, to decode stimulus conditions with two dimensions, one should at least include three PCs. If the authors can include three PCs in the comparison and still show better performance with the new approach, it will make the conclusion stronger.

2. Demixed PCA has been growing popular in recent years. It is a similar approach to the proposed method. I would appreciate it if the authors can describe the differences, or even make a comparison, between the two approaches.

Minor concerns:

1. Page 4. Last paragraph. The authors should add the letter of panels in Fig 2. Same for the text related to Figure 4.

2. Figure 4a. What is the x-axis?

3. Figure 5g. Why was the performance of taPCA reduced when the trial number increased?

Reviewer #2: In the manuscript " Dimensionality reduction for neural population decoding", Heller et al. report a new method for dimensionality reduction of neural population data. This approach projects high-dimensional neural activity into a two-dimensional space by capturing the variance of stimulus-evoked activity (signal axis) and the stimulus-independent trial to trial variability (noise axis) separately. It shows a significant advantage over standard principal component analysis in stimulus discriminations, especially in conditions with a fewer number of observations. The outcome is easy to interpret since it visualizes the signal and noise information separately in a 2-D space. Although the approach is limited by only working in a pairwise way and only capturing the 1st primary dimension of noise correlation variability, it is still a simple but effective method that could serve as an alternative approach to decoding analysis with fewer observations. The approach could be of interest to the field, and I would recommend the publication of it in PLOS One.

6. PLOS authors have the option to publish the peer review history of their article (what does this mean?). If published, this will include your full peer review and any attached files.

Reviewer #1: No

Reviewer #2: **Yes: **Bruno Averbeck

---

## [Author Response · Author response to Decision Letter 0]

31 May 2022

We have included a document, "Response to Reviewers.pdf" that addresses all specific editor and reviewer comments. The text of the document is also pasted below, if needed.

Thank you for the positive feedback on our manuscript “Targeted dimensionality reduction enables reliable estimation of neural population coding accuracy from trial-limited data''. We have addressed all points contained in the decision letter and outlined the details below. 

Editor comments / journal requirements 

We have made all necessary changes to comply with journal requirements for formatting and file naming. These changes are reflected in the submitted documents. 

2. Please amend either the title on the online submission form (via Edit Submission) or the title in the manuscript so that they are identical. 

The title of the manuscript has been updated to match the full title included in the online submission form. 

“This work was supported by a National Science Foundation Graduate Research Fellowship (NSF GRFP, GVPRS0015A2) (CRH), the National Institute of Health (NIH, R01 DC0495) (SVD), Achievement Rewards for College Scientists (ARCS) Portland chapter (CRH), and by the Tartar Trust at Oregon Health and Science University (CRH).” 

“This work was supported by a National Science Foundation Graduate Research Fellowship (NSF GRFP, GVPRS0015A2) (CRH), the National Institute of Health (NIH, R01 DC0495) (SVD), Achievement Rewards for College Scientists (ARCS) Portland chapter (CRH), and by the Tartar Trust at Oregon Health and Science University (CRH). The funders had no role in study design, data collection and analysis, decision to publish, or preparation of the manuscript.” 

We have removed the funding statement from our manuscript. The funding statement above is correct. There is no need to amend it. 

We have included the following ethics statement in our Methods section (lines 46-49) in the revised manuscript): 

“All procedures were performed in accordance with the Oregon Health and Science University Institutional Animal Care and Use Committee (IACUC) and conform to standards of the Association for Assessment and Accreditation of Laboratory Animal Care (AAALAC).” 

Please let us know if this needs to be changed in any way. 

Reviewer comments 

We appreciate the reviewers’ positive and thoughtful comments. We have revised the manuscript in a way that we hope addresses their concerns. A detailed response to each specific point is provided below. We have also included a marked-up copy of the manuscript that shows all changes from the previous version. 

Reviewer #1: 

In the current work, the authors proposed a novel computation approach to decode high-dimensional neural data. In this approach, the authors proposed to use the mean of different experimental conditions and the first eigenvector of the population covariance to determine the decoding axis, which has the potential to achieve better decoding performance with limited trials than the traditional PCA. The work is technically solid and interesting. 

A major issue is the very abbreviated description of the rationale and technical details. For example, the first section of Results described some very basic concepts of this approach. But for anyone who has not read Averbeck and Lee 2006, it would be very hard to understand. 

We agree that there were important points that required further clarification. In particular, a more careful treatment of the rationale behind our method was missing. For anyone not familiar with the aforementioned work, the terminology and concepts could be challenging. Therefore, we have added a new section at the beginning of the Results, “Neural population decoding and noise correlations,” in which we introduce the main principles that form the conceptual basis for our work. We define the terms “noise correlation” and “state space” and discuss neural decoding conceptually before diving more deeply into the math in the subsequent results. We also revised the remaining text in several places to ensure that we are consistent in our use of this terminology. Finally, we added a new figure (Fig. 1 in the revision), adapted from Averbeck & Lee, that graphically illustrates the conceptual basis that this work builds on. We hope these changes help to orient readers sufficiently before describing our new method. 

The new results section is located at lines 114-156 in the revised manuscript. 

Other major concerns: 

1. To show the advantage of the new method over traditional PCA, authors applied each method to neural data collected from auditory cortices. The general result did show a clear advantage of the new method over PCA. However, in this comparison, only the first two PCs were used to decode stimulus conditions that have two dimensions. In analyses of auditory neural data, the first PC of PCA usually reflects the rise and fall of the sound, while the second and third PCs contain information about stimulus conditions. Therefore, to decode stimulus conditions with two dimensions, one should at least include three PCs. If the authors can include three PCs in the comparison and still show better performance with the new approach, it will make the conclusion stronger. 

We would like to thank the reviewer for this insightful comment and offer some additional clarification. In the analyses presented in this manuscript, we collapsed the sound evoked neural response for each stimulus into a single time bin, so the dynamics of the response were not considered in this example. We have added the following sentence to the final results section to help clarify this point: 

“The neural response to each stimulus was defined as the total number of spikes detected during the 300 ms stimulus duration for each neuron.” (lines 316 – 318) 

Therefore, adding additional PCs cannot provide any more information about the stimulus condition. We made the choice to collapse activity over time to keep our analysis as simple and general as possible. We concede that this is a simplified approach and that we are likely missing important time-varying, sound-evoked activity. However, our method can easily be adapted to treat each different time point as its own stimulus condition – in other words, a separate dDR decomposition could be performed for each time bin. This would allow one to evaluate how optimal sound decoding changes as a function of time from e.g., stimulus onset. As response dynamics were not the focus of this manuscript, per se, we did not include this analysis. 

2. Demixed PCA has been growing popular in recent years. It is a similar approach to the proposed method. I would appreciate it if the authors can describe the differences, or even make a comparison, between the two approaches. 

This is an important point that we neglected to fully address in the original manuscript. Thank you for bringing it to our attention. There are at least two critical distinctions between our method and dPCA. 

First, our method is strictly focused on optimal decoding. It should be thought of as a preprocessing, dimensionality reduction, step prior to applying a method such as Linear Discriminant Analysis (LDA). By using dDR before LDA, one can mitigate overfitting issues that make application of standard LDA to single-unit population data challenging in practice. While one goal of dPCA is indeed to provide a low dimensional representation for decoding, another goal is to maintain a faithful representation of the true geometric structure of the data. This latter goal does not need to be compatible with optimal decoding; therefore, the low-D projection found with dPCA will not necessarily provide optimal decoding of experimental conditions. This trade-off is illustrated very nicely in Figure 2 of Kobak et al., 2016. 

Second, our method is developed for pairwise experimental conditions. That is, a different dDR projection exists for each pair of stimuli. dPCA is useful when there are more than two. In particular, dPCA can provide a useful tool for marginalization when the data spans multiple dimensions (e.g., stimulus X time X neurons or stim X behavioral state X neurons) and an interpretable, constant low-D space across all dimensions is desired. 

To address this point, we have added a section to the Discussion, titled “Related methods for dimensionality reduction and neural decoding,” which reviews the key differences between our method and other related approaches for dimensionality reduction and neural decoding, namely standard PCA, LDA, and dPCA. This is located at lines 418-451 in the revised text. 

Minor concerns: 

1. Page 4. Last paragraph. The authors should add the letter of panels in Fig 2. Same for the text related to Figure 4. 

We have edited the text to specify figure panels in all cases where a figure is cited. These changes are reflected in the marked-up copy of our revised manuscript. 

2. Figure 4a. What is the x-axis? 

We believe the reviewer is referring to Fig 5a (now 6a, after revision), where the x-axis label was omitted and is shared with the panel below (Fig 6b). We have added a label to the x-axis in panel a (“Noise Center Frequency”) and revised the figure legend to explicitly state this as well. 

3. Figure 5g. Why was the performance of taPCA reduced when the trial number increased? 

Thank you for noticing this. There was a mistake in the way we were calculating our bootstrapped estimates of standard error for each sample size. Briefly, we did not correctly control how we randomly resampled our dataset for cross-validated estimates of decoding accuracy. This introduced variability between the data included in the different sample sizes, leading to a spurious apparent drop in decoding performance for taPCA at high repetition counts. This error did not affect any of the conclusions we present in the manuscript, but did make the results confusing. 

We have corrected this mistake and added a section to the Materials and Methods titled “Bootstrapped estimates of decoding performance for different sample sizes” (lines 91 - 112). Here we describe the bootstrapping procedure used for estimating decoding performance across different sample sizes in detail. Additionally, we have modified the statistic plotted on the y-axis in Fig. 6 panels d and g to more clearly demonstrate the benefit of dDR over PCA methods. Previously, we reported absolute decoding performance for each method in units of d’2. To more directly compare the methods, we now instead report performance of PCA methods as a fraction of dDR performance. We believe this unit-less quantity provides a more interpretable and direct illustration of the benefit of dDR when applied to real neural data. These changes are reflected in Fig. 6 of the revised manuscript. 

Reviewer #2: 

In the manuscript " Dimensionality reduction for neural population decoding", Heller et al. report a new method for dimensionality reduction of neural population data. This approach projects high-dimensional neural activity into a two-dimensional space by capturing the variance of stimulus-evoked activity (signal axis) and the stimulus-independent trial to trial variability (noise axis) separately. It shows a significant advantage over standard principal component analysis in stimulus discriminations, especially in conditions with a fewer number of observations. The outcome is easy to interpret since it visualizes the signal and noise information separately in a 2-D space. Although the approach is limited by only working in a pairwise way and only capturing the 1st primary dimension of noise correlation variability, it is still a simple but effective method that could serve as an alternative approach to decoding analysis with fewer observations. The approach could be of interest to the field, and I would recommend the publication of it in PLOS One. 

We thank the reviewer for these supportive comments.

---

## [Decision Letter · Decision Letter 1]

24 Jun 2022

Targeted dimensionality reduction enables reliable estimation of neural population coding accuracy from trial-limited data

PONE-D-22-02203R1

Dear Dr. Heller,

We’re pleased to inform you that your manuscript has been judged scientifically suitable for publication and will be formally accepted for publication once it meets all outstanding technical requirements.

Thank you for your thorough response to the reviewer's comments.

Kind regards,

David S Vicario, Ph.D.

Academic Editor

PLOS ONE

Additional Editor Comments (optional):

Reviewers' comments:

Reviewer's Responses to Questions

**Comments to the Author**

1. If the authors have adequately addressed your comments raised in a previous round of review and you feel that this manuscript is now acceptable for publication, you may indicate that here to bypass the “Comments to the Author” section, enter your conflict of interest statement in the “Confidential to Editor” section, and submit your "Accept" recommendation.

Reviewer #1: All comments have been addressed

Reviewer #2: All comments have been addressed

2. Is the manuscript technically sound, and do the data support the conclusions?

Reviewer #1: Yes

Reviewer #2: Yes

3. Has the statistical analysis been performed appropriately and rigorously? 

Reviewer #1: Yes

Reviewer #2: Yes

4. Have the authors made all data underlying the findings in their manuscript fully available?

Reviewer #1: Yes

Reviewer #2: Yes

5. Is the manuscript presented in an intelligible fashion and written in standard English?

Reviewer #1: Yes

Reviewer #2: Yes

6. Review Comments to the Author

Reviewer #1: The revised version is significantly improved. All reviewers’ comments were properly addressed. The authors added an extra section to describe the rationale of the work. The example with the dDR analysis was also improved and more convincing. I also appreciate that the authors compared the new approach with other similar techniques in the discussion. The article is clear and much easier to read now. I would recommend it for publication.

Reviewer #2: The authors have addressed the main comments. I am happy to recommend this manuscript for publication.

7. PLOS authors have the option to publish the peer review history of their article (what does this mean?). If published, this will include your full peer review and any attached files.

Reviewer #1: No

Reviewer #2: **Yes: **Bruno B. Averbeck

---

## [Editor Report · Acceptance letter]

28 Jun 2022

PONE-D-22-02203R1 

Targeted dimensionality reduction enables reliable estimation of neural population coding accuracy from trial-limited data 

Dear Dr. Heller:

I'm pleased to inform you that your manuscript has been deemed suitable for publication in PLOS ONE. Congratulations! Your manuscript is now with our production department. 

Kind regards, 

on behalf of

Dr. David S Vicario 

Academic Editor

PLOS ONE